# Severity of COVID-19 among Hospitalized Patients: Omicron Remains a Severe Threat for Immunocompromised Hosts

**DOI:** 10.3390/v14122736

**Published:** 2022-12-08

**Authors:** Louis Nevejan, Sien Ombelet, Lies Laenen, Els Keyaerts, Thomas Demuyser, Lucie Seyler, Oriane Soetens, Els Van Nedervelde, Reinout Naesens, Dieter Geysels, Walter Verstrepen, Lien Cattoir, Steven Martens, Charlotte Michel, Elise Mathieu, Marijke Reynders, Anton Evenepoel, Jorn Hellemans, Merijn Vanhee, Koen Magerman, Justine Maes, Veerle Matheeussen, Hélène Boogaerts, Katrien Lagrou, Lize Cuypers, Emmanuel André

**Affiliations:** 1Department of Laboratory Medicine, National Reference Center for Respiratory Pathogens, UZ Leuven—University Hospitals Leuven, Herestraat 49, 3000 Leuven, Belgium; 2KU Leuven Department of Microbiology, Immunology and Transplantation, Laboratory of Clinical Microbiology, 3000 Leuven, Belgium; 3Department of Microbiology and Infection Control, Vrije Universiteit Brussel (VUB), UZ Brussel—University Hospitals Brussels, 1090 Brussels, Belgium; 4Center for Neurosciences, Faculty of Medicine and Pharmacy, Vrije Universiteit Brussel (VUB), 1090 Brussels, Belgium; 5Department of Internal Medicine and Infectious Diseases, Vrije Universiteit Brussel (VUB), UZ Brussel—University Hospitals Brussels, 1090 Brussels, Belgium; 6Department of Medical Microbiology, Department of infection prevention and control, ZNA Middelheim, 2020 Antwerp, Belgium; 7Clinical Laboratory of Microbiology, OLV Hospital, 9300 Aalst, Belgium; 8Department of Microbiology, Laboratoire Hospitalier Universitaire de Bruxelles, Université Libre de Bruxelles, 1000 Brussels, Belgium; 9Department of Laboratory Medicine—Medical Microbiology, AZ Sint Jan Brugge-Oostende, 8000 Brugge, Belgium; 10Clinical Laboratory, Jessa Hospital, 3500 Hasselt, Belgium; 11Department of Microbiology, University Hospital Antwerp, 2650 Antwerp, Belgium

**Keywords:** COVID-19, omicron, delta, immunocompromised host, hospitalization

## Abstract

The Omicron variant of severe acute respiratory syndrome coronavirus 2 (SARS-CoV-2) emerged in the general population in the context of a relatively high immunity gained through the early waves of coronavirus disease 19 (COVID-19), and vaccination campaigns. Despite this context, a significant number of patients were hospitalized, and identifying the risk factors associated with severe disease in the Omicron era is critical for targeting further preventive, and curative interventions. We retrospectively analyzed the individual medical records of 1501 SARS-CoV-2 positive hospitalized patients between 13 December 2021, and 13 February 2022, in Belgium, of which 187 (12.5%) were infected with Delta, and 1036 (69.0%) with Omicron. Unvaccinated adults showed an increased risk of moderate/severe/critical/fatal COVID-19 (crude OR 1.54; 95% CI 1.09–2.16) compared to vaccinated patients, whether infected with Omicron or Delta. In adults infected with Omicron and moderate/severe/critical/fatal COVID-19 (*n* = 323), immunocompromised patients showed an increased risk of in-hospital mortality related to COVID-19 (adjusted OR 2.42; 95% CI 1.39–4.22), compared to non-immunocompromised patients. The upcoming impact of the pandemic will be defined by evolving viral variants, and the immune system status of the population. The observations support that, in the context of an intrinsically less virulent variant, vaccination and underlying patient immunity remain the main drivers of severe disease.

## 1. Introduction

Compared to the ancestral severe acute respiratory syndrome coronavirus 2 (SARS-CoV-2) Wuhan-Hu-1 strain and to the SARS-CoV-2 lineage B.1.617.2 (Delta variant), the SARS-CoV-2 lineage B.1.1.529 (Omicron variant) is characterized by an antigenic drift of 30, respectively, 28 mutations in the viral spike protein [1]. As a result, Omicron showed more ability than Delta to cause infections in individuals with acquired immunity from either prior infection or vaccination [2]. Omicron outcompeted the Delta variant almost completely in only four weeks in the area of detected emergence, South Africa [3], and caused a nearly complete viral population replacement in a matter of weeks with >3 million daily new confirmed cases worldwide by mid-January 2022 [4]. Initially, the Omicron variant was divided into sub-lineages aliased BA.1 (the main clade), BA.2, and BA.3 [2]. Two distinct sub-lineages BA.4 and BA.5 were designated later on [5].

Increased transmissibility had already been observed with previous variants in the two-year coronavirus disease 2019 (COVID-19) pandemic. The emergence of the SARS-CoV-2 lineage B.1.1.7 (Alpha variant), first detected in the United Kingdom (UK) in September 2020, was associated with a higher transmissibility, but also with an increased severity of disease [6] and risk of death [7], compared to the first COVID-19 pandemic wave. In March 2021, the Delta variant rapidly spread in the UK and showed an even higher risk of hospital admission due to COVID-19 in the unvaccinated population, compared to the patients infected with the Alpha variant [8]. The higher intrinsic severity of the preceding SARS-CoV-2 variants with enhanced transmissibility, compared to the previous ancestor or previous dominant variant, urged studies to investigate the severity of COVID-19 caused by the even more infective Omicron variant [9].

However, early data from South Africa were encouraging. Patients infected with the Omicron variant showed a reduced risk of hospitalization, a shorter length of hospital stay [10], and reduced risk of severe disease compared to patients infected with the Delta variant [3], but also compared to patients infected with the ancestral strain (with D614G mutation), and patients infected with the SARS-CoV-2 lineage B.1.351 (Beta variant) [11,12]. This less severe disease remained after adjusting for both vaccination and prior diagnosed infection indicating a possible intrinsically reduced virulence of Omicron [13].

Extrapolations of the South African studies to other populations should be carried out with caution. Indeed, the age distribution may be different, with a predominantly younger population in South Africa. The vaccination status may also differ, with +/− 35% of the adult population in South Africa having received a basic COVID-19 vaccination (messenger RNA (mRNA) or viral vector vaccine) at the time of Omicron’s first detection. In addition, South Africa had high levels of prior infections, with about 70% unvaccinated patients in the province of Gauteng showing SARS-CoV-2 anti-S or anti-N IgG seropositivity at that time [11,12]. Yet, preliminary data retrieved the during the first Omicron emergence from Europe [14,15], the UK [16] and North America [17,18,19,20] confirmed these early observations of reduced risk of hospitalization, shorter duration of hospital stay, and fewer severe outcomes, despite an increased incidence rate of Omicron infections.

Despite the overall lower virulence of Omicron, the identification of patient groups remaining vulnerable to severe COVID-19 and prolonged hospitalizations is important to guide health policies when new waves of infections emerge. Data on COVID-19 outcomes, obtained during previous waves in mainly unvaccinated and unexposed patients, showed an association between severity of COVID-19 and increasing age, masculine gender, and a wide range of pre-existing medical conditions, such as diabetes, obesity, heart failure, renal disease, chronic obstructive pulmonary disease (COPD), and immunocompromised status [21]. However, with the emergence of Omicron, evaluating COVID-19 severity is more challenging, given vaccination coverage and a global exposition to previous variants.

It, therefore, seemed interesting to look at the situation in Belgium, characterized by a high vaccination coverage at the time of Omicron’s first infection: 88% of adults were fully vaccinated, and 17% of adults had already received a first booster dose [22]. In this multicenter retrospective study, we compared hospitalized patients infected with the Delta variant and with the Omicron variant looking at COVID-19 severity and short-term clinical outcome, taking vaccination status and immune status into account.

## 2. Materials and Methods

All hospitalized patients with a PCR confirmed SARS-CoV-2 infection from four university and four general hospitals in Belgium were included retrospectively, if admitted between 13 December 2021, and 13 February 2022 (*n* = 9 weeks). During this period, all newly hospitalized patients were screened for SARS-CoV-2 positivity, regardless of COVID-19 symptoms, in all participating centers in accordance with national testing guidelines at that time. This period was characterized by the full viral replacement of Delta by Omicron BA.1, followed by the emergence of Omicron BA.2, causing a fifth wave of COVID-19 hospital admissions in Belgium, with the highest new number of hospital admissions on 2 February 2022 (*n* = 2576) [4,23]. Exclusion criteria were admission to the emergency department solely; to a psychiatric or rehabilitation ward; admission to ambulatory care/day surgery; or a positive SARS-CoV-2 screening result, interpreted as a viral remnant of a previous infection (based on viral load <3.0 log copies/mL and/or clinical context).

The following demographic and clinical information were collected: age at admission, sex, date of hospitalization, ward of hospitalization (intensive care unit (ICU) or non-ICU including potential date of transfer to ICU), and mortality during hospitalization (patients still hospitalized on 3 May 2022 were considered non-deceased). COVID-19 symptoms at time of hospital admission (based on National Institutes of Health (NIH) classification [24] (Appendix A)) were registered, as well as the most severe COVID-19 symptoms during hospitalization. If available, SARS-CoV-2 serology data before or during the first days of admission were collected (SARS-CoV-2 IgG anti-S and/or anti-N). The immune status of patients at time of admission, according to CDC criteria [25], were registered, and immunocompromised status was defined as (1) receiving active cancer treatment for tumors or cancers of the blood; (2) receiving immunosuppressants after organ transplant; (3) recipient of a stem cell transplant within the last 2 years or taking immunosuppressants; (4) moderate or severe primary immunodeficiency; (5) advanced or untreated HIV infection; (6) receiving high-dose corticosteroids (Appendix A). COVID-19 vaccination status (including date of last vaccine) of the patient on admission was registered, further [25] described in Appendix A. Regarding SARS-CoV-2 screening results, both the viral load and the variant-of-concern (VOC) were registered, with typing of the VOC being based on whole-genome sequencing results (divided into Delta variant, Omicron BA.1 variant, Omicron BA.1.1 variant, Omicron BA.2 variant, Omicron BA.3 variant, other variant or untypable) or based on a genotyping PCR when WGS could not be performed. Previous SARS-CoV-2 infections were registered if detected in the same hospital as the current admission. An overview of the SARS-CoV-2 serology, PCR and WGS assays, and analyzers is available in Appendix A.

This study was approved by the Ethics Committee Research UZ/KU Leuven (reference S66037, date of approval 28 February 2022). After signing a Data Transfer Agreement for non-commercial studies, the participating centers sent pseudonymized patients’ data to the principal investigator via a secured Excel file. Data were processed using Microsoft^®^ Excel^®^ Version 2204 (Microsoft, Redmond, WA, USA) and R 4.1.3 (RStudio version 2022.02.1) [26].

Statistical analyses comprised bivariate analyses, and multivariate logistic regression analyses. Bivariate analyses were performed by using a Chi-squared test in case of categorical data (reported as absolute number, *n*, and relative frequency, %). Continuous variables (reported as median and interquartile range (IQR)) were compared using Wilcoxon rank-sum test (sample size ≤ 30 and non-paired non-normally distributed data), or two-sample, unpaired *t*-test (sample size ≤ 30 and normally distributed data or sample size > 30). Multivariable logistic regression analysis was performed to assess the effect of independent variables on binary outcomes, such as severity of disease during hospitalization (asymptomatic or mild versus moderate, severe, critical or fatal COVID-19), mortality (discharged or deceased not due to COVID-19 versus deceased due to COVID-19), and need for ICU admission (admission to a non-ICU ward or admission to ICU not due to COVID-19 versus ICU admission due to COVID-19). Independent variables included age at admission, sex, VOC (Delta versus Omicron), immune status at admission (immunocompromised versus non-immunocompromised), vaccination status (unvaccinated versus vaccinated; non-boosted versus boosted; viral vector versus mRNA vaccines) and time since last vaccination (<2/4/6 months before admission versus >2/4/6 months before admission). Because of the multicenter nature of the study, the variable ‘hospital’ was included as a random effect, using a mixed-effects logistic regression model (lme4 package in R). However, due to high numbers of missing data regarding vaccination data, neither multivariable nor mixed-effects models could be run; a bivariate regression model was used instead for these variables. All *p*-values <0.05 were considered statistically significant.

## 3. Results

### 3.1. Patient Demographics

Between 13 December 2021, and 13 February 2022, 1501 newly hospitalized patients with positive SARS-CoV-2 test result were included. In total, 187 patients (12.5%) were infected with Delta, 1036 patients (69.0%) with Omicron, in 278 patients (18.5%) no VOC could be typed (Appendix A). Clinical characteristics of all patients are shown in Appendix A. No or mild COVID-19 symptoms were developed in 160/210 (76.2%) children and 753/1291 (58.3%) adults during hospitalization (Appendix A). ICU hospitalization accounted for 20/210 (9.5%) children and 178/1291 (13.8%) adults of which, respectively, 8/20 (40.0%) and 49/178 (27.5%) had no or mild COVID-19 symptoms. No children, and 139/1291 (10.8%) adult patients died during hospitalization, however, 40/139 (29%) of them from non-COVID-19 related causes.

### 3.2. Hospitalized Adults Infected with Omicron Showed a Reduced Intrinsic COVID-19 Severity Compared to Those Infected with Delta

Looking at adult patients with known VOC (Delta and Omicron together, *n* = 1071/1291, 83.0%), 449/1071 (41.9%) suffered from moderate/severe/critical/fatal COVID-19. Patients infected with Omicron had reduced odds to suffer from these symptoms, compared to patients infected with Delta, after correcting for age, sex, and immune status (odds ratio (OR) 0.14; 95% confidence interval (CI) 0.09–0.22; *p* < 0.001) (Table 1, Figure 1A). Next, the risk of ICU admission due to COVID-19 (total on ICU, *n* = 112/1071, 10.5%) was reduced for patients with Omicron infection, compared to patients infected with Delta, corrected for using the same variables as above (OR 0.26, 95% CI 0.17–0.40; *p* < 0.001). Finally, regarding the risk of in-hospital mortality related to COVID-19 (total mortality, *n* = 89/1071, 8.3%), patients infected with Omicron showed reduced odds, compared to patients infected with Delta, corrected for the same variables as above (OR 0.24; 95% CI 0.14–0.40; *p* < 0.001).

Using bivariate logistic regression, unvaccinated patients were more likely than vaccinated patients ([1/2] basic or boosted vaccination scheme) to develop moderate/severe/critical/fatal COVID-19 (crude OR 1.54; 95% CI 1.09–2.16; *p* = 0.01). Similarly, recently vaccinated (<2 months before admission) were less likely than patients vaccinated >2 months before admission to develop moderate/severe/critical/fatal COVID-19 (crude OR 0.55; 95% CI 0.38–0.78; *p* = 0.001). Moreover, unvaccinated patients showed increased odds, compared to vaccinated patients, for admission to ICU due to COVID-19 (crude OR 1.80; 95% CI 1.10–2.87; *p* = 0.02). No significant differences were noticed in vaccination status between patients deceased due to COVID-19 and patients discharged or deceased not due to COVID-19 (crude OR 1.15; 95% CI 0.56–2.19; *p* = 0.68, unadjusted for age, sex, and immune status) (Table 1).

A similar comparative analysis in children with known VOC (*n* = 152/210, 72.4%) was not possible due to a low number of Delta-infected patients (Appendix A).

### 3.3. Immunocompromised Adults HOSPITALIZED with Omicron Infections Had an Increased Risk of Severe COVID-19 Outcomes

Looking at the clinical course of Delta-infected adults separately (*n* = 166), immunocompromised status was not significantly associated with the development of moderate/severe/critical/fatal COVID-19, after correcting for age and sex; nor the need for ICU admission; nor in-hospital mortality related to COVID-19 (Appendix A). However, in Omicron-infected adults (*n* = 905), both an older age, male sex, and immunocompromised status were associated with significantly increased odds for moderate/severe/critical/fatal COVID-19, and in-hospital mortality related to COVID-19 (Appendix A). Within the Omicron subgroup, recent vaccination (<2 months before admission) was associated with lower odds for moderate/severe/critical/fatal COVID-19 symptoms (crude OR 0.57; 95% CI 0.38–0.82; *p* = 0.003; unadjusted for age, sex, immune status). Neither ICU admission nor in-hospital mortality related to COVID-19 were significantly associated with vaccination in bivariate analysis in the Omicron subgroup (Appendix A).

Next, COVID-19 outcomes were investigated in Omicron-infected adults with moderate/severe/critical/fatal COVID-19 only (*n* = 323) i.e., excluding patients with asymptomatic/mild symptoms. As shown in Figure 1B and Appendix A, immunocompromised patients had a significantly increased risk of all defined outcomes, i.e., risk of critical/fatal COVID-19 (OR 2.42; 95% CI 1.39–4.22; *p* = 0.002); risk of admission to ICU due to COVID-19 (OR 1.96; 95% CI 1.06–3.57; *p* = 0.03); and risk of in-hospital mortality related to COVID-19 (OR 3.13; 95% CI 1.52–6.62; *p* = 0.002), all corrected for age and sex. Neither the vaccination status—unadjusted for age, sex, immune status (vaccinated versus unvaccinated, boosted versus non-boosted, mRNA vaccination versus viral vector vaccination)—nor the time since last vaccination (<2, <4 or <6 months before admission) showed a significant effect on the defined outcomes using bivariate analysis (Appendix A).

### 3.4. Neither Vaccination Status Nor Immune Status Determine Total Hospital Length-of-Stay in Omicron-Infected Adults

Within the subgroup of Omicron-infected adults with moderate/severe/critical/fatal COVID-19 solely i.e., excluding patients with asymptomatic/mild symptoms, no significant association was found between the patients’ vaccination status or immune status, and the total hospital length-of-stay. Moreover, when comparing total hospital length-of-stay between patients with different vaccination status or immune status, regardless of the category of symptoms, no difference was noticed (Appendix A).

## 4. Discussion

In this multicenter observational cohort study, we investigated the clinical differences between patients infected with the SARS-CoV-2 Delta variant and patients infected with the SARS-CoV-2 Omicron variant in hospitalized patients. We showed that patients infected with Omicron had significantly reduced odds (1) to develop moderate/severe/critical/fatal COVID-19, (2) for COVID-19 associated ICU admission, and (3) for in-hospital mortality related to COVID-19, compared to patients infected with Delta (corrected for age, sex, and immune status at admission). These data confirm previous reports of an inherently lower severity of Omicron, compared to Delta, resulting in a reduced risk of hospitalization [3,14,16,27,28], less severe illness [11,17,18,20,29], and a lower case-fatality ratio [11,19].

However, as the Delta variant is no longer circulating [23], the important considerations for potential COVID-19 waves in the future—with Omicron or another VOC—are the identification of groups requiring additional protection against infection, and severe outcomes. First, our data confirmed the crucial importance of vaccination. With substantially higher vaccination rates in the West, compared to South Africa, at the time of emergence of Omicron (+/− 60% in European and North American versus +/− 30% in South African adults [4]), it has been clearly shown that unvaccinated Omicron-infected patients have a higher risk of hospitalization compared to patients who received two or three vaccine doses [27]. Although our study did not investigate the actual risk of hospitalization, we could show that unvaccinated adult patients have a higher risk of moderate/severe/critical/fatal COVID-19 (OR 1.54; 95% CI 1.09–2.16) and of COVID-19 associated mortality (OR 1.80; 95% CI 1.10–2.87) compared to vaccinated patients (Table 1). Additionally, looking at the subgroup of Omicron-infected adult patients (Appendix A), we found a reduced risk of moderate/severe/critical/fatal COVID-19 of recently vaccinated patients (<2 months before admission) (crude OR 0.57; 95% CI 0.38–0.82). However, in bivariate analysis, no difference in need for ICU admission nor risk of in-hospital mortality was found between vaccinated and unvaccinated patients in the Omicron subgroup. This could be explained by the lack of adjustment for age—due to low sample size—but also by the lack of reinfection data, since unvaccinated patients with earlier documented SARS-CoV-2 infection gain moderate protection against COVID-19 associated hospitalization [28].

Secondly, immunocompromised adult patients showed an increased risk of moderate/severe/critical/fatal COVID-19, ICU admission, and in-hospital mortality related to COVID-19, compared to non-immunocompromised patients (Table 1). This risk factor was only significant in Omicron-infected adults, not in Delta-infected adults (however, the statistical power in Delta-infected adults was limited due to a smaller sample size). In addition, excluding asymptomatic/mild infections caused by Omicron, immunocompromised adult patients showed an increased risk of critical/fatal COVID-19 (Appendix A). A possible underreporting of immunocompromised patients infected with Delta cannot be ruled out, since data were collected during a higher circulation of Omicron in the general community, causing an increased exposure of the virus in this specific patient group. Additionally, considering the higher intrinsic virulence of Delta, more non-immunocompromised patients were at risk of severe outcomes during these Delta waves. Studies on the hospitalization course of COVID-19 in immunocompromised patients are scarce. Mahale et al. [30] showed a high requirement of hospital admission (20%) with substantial morbidity in 114 immunocompromised patients, despite a high vaccination coverage in these patients. Our data confirm that despite a decreased intrinsic virulence of Omicron, humoral and cellular immune responses after vaccination in these patients offer insufficient protection against the most severe COVID-19 outcomes, requiring additional protection measurements (e.g., monoclonal antibodies, directly acting antiviral drugs, modified COVID-19 vaccines) [30,31,32,33].

Considering the apparent decrease in virulence of the most recent SARS-CoV-2 VOC, it remains of utmost importance to continue monitoring clinical outcomes in different patient populations, especially when new VOCs emerge. The combination of high incidence rates and lower virulence of Omicron highlights the importance of distinctive reporting of incidental detection of asymptomatic SARS-CoV-2 infections, and symptomatic COVID-19 infections. Our data showed that 160/210 (76.2%) hospitalized children and 753/1291 (58.3%) hospitalized adults experienced no or only mild COVID-19 symptoms; thus concluding that these patients were admitted for other pathologies that were not COVID-19 (nearly identical numbers as reported in South Africa [10]). When looking at patients admitted to ICU, 8/20 (40%) children and 49/178 (28%) adults were not admitted to ICU for COVID-19 symptoms. In the current situation of the pandemic, simultaneous registration of hospital occupancy, and of concise clinical information, is warranted to organize hospital wards adequately (need for exclusive COVID-19 wards versus admitting SARS-CoV-2 positive patients to other pathology-specific wards) [29,34].

By reviewing clinical data using individual patient records, strengths of our study include an accurate classification of a large cohort of patients (*n* = 1501) according to COVID-19 symptoms. In addition, this is one of the largest studies showing the increased vulnerability of immunocompromised hospitalized patients when infected with Omicron. The limitations of this multicenter study must be addressed too.

First, some crucial clinical information was lacking during data collection. Documented previous infections were highly underreported caused by an inaccessibility to the national SARS-CoV-2 sampling database. However, the emergence of Omicron was associated with a significant increase in reinfection cases [29,35]. Therefore, the effects of acquired immunity due to previous infection could not be included in the multivariate analyses. Other important missing data were due to inaccessibility of the national vaccination database of deceased patients, leading to unknown vaccination status for deceased patients when the vaccination data were not explicitly mentioned in the medical files (i.e., in 30/99 (30.3%) patients deceased from COVID-19, vaccination status was unknown). Additionally, data on general comorbidities (e.g., diabetes, renal insufficiency) were not collected. However, immunocompromised patients very often have additional comorbidities (e.g., steroids-induced diabetes, chronic kidney disease in nonrenal organ transplant recipients), so the exact attributable risk of being immunocompromised could potentially be outweighed in the current analysis. Next, data on the specific subgroup of immunocompromised status (according to CDC criteria [25]) were not collected, hampering additional sub-analyses in these patient groups. It should be noted that CDC criteria defining immunocompromised patients are very broad, as it has been recently reported that seroconversion after a booster vaccine is not equal between patients with solid organ malignancies receiving recent systemic anticancer therapy, and patients with hematological malignancies receiving B-cell depleting therapies [36].

Second, assigning COVID-19 patients into the classification of symptoms we used was not always straightforward. For example, a geriatric patient admitted with general deterioration with positive SARS-CoV-2 screening but no clinical nor radiological evidence of a lower respiratory infection was classified in the ‘mild symptoms’ category, even if residual viral infection might have contributed to her hospitalization. In contrast, in SARS-CoV-2 positive children with severe respiratory symptoms and concomitant respiratory syncytial virus (RSV) infection, the causative role for SARS-CoV-2 is questionable. In general, the lack of clinical information about potential respiratory co-infections, in both adults and children, could have resulted in a biased classification of symptoms. Additionally, the number of children infected with Omicron and with severe COVID-19 outcomes was too low for further analyses.

Third, data collection started at a time of decreasing viral circulation of the Delta variant, at the ‘tail’ of infections and hospitalization of patients infected with the Delta variant, compared to the Omicron variant. However, this was also a deliberate choice to minimize differences in vaccination status between the two patient groups. Indeed, a lower vaccination coverage of hospitalized patients in the previous Delta waves could potentially introduce bias, when comparing their outcomes with those of a higher vaccinated population during the Omicron wave. Lastly, the use of monoclonal antibodies was not registered in the database, and this may be of importance when comparing clinical outcomes in immunocompromised patients, with or without previous treatment with monoclonal antibodies.

## 5. Conclusions

The antibody-evasive nature of the Omicron SARS-CoV-2 variant has led to a worldwide increased incidence of COVID-19, and rapid displacement of the Delta variant. In this multicenter study, the lower intrinsic virulence of Omicron was demonstrated by reduced odds of moderate/severe/critical/fatal COVID-19, need for ICU admission, and COVID-19 associated in-hospital mortality in Omicron-infected patients, compared to patients infected with Delta. Additionally, of importance was the high number of hospitalized children and adults with no or solely mild COVID-19 symptoms (respectively, 76.2% and 58.3%). This highlights the need for a symptomatic registration of hospitalized COVID-19 patients to plan healthcare capacity needs adequately, and more efficiently. In symptomatic Omicron-infected patients, immunocompromised patients had a significantly increased risk of suffering from the most severe COVID-19 outcomes. Despite the lower intrinsic virulence of Omicron, additional precautions are still warranted to protect the vulnerable immunocompromised patients especially when viral transmission in the community is high.

## Figures and Tables

**Figure 1 viruses-14-02736-f001:**
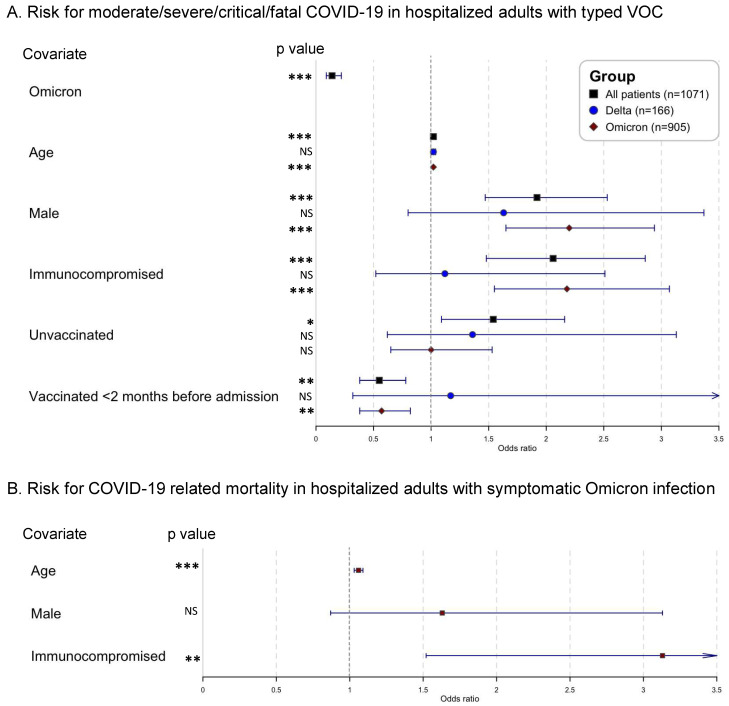
Forest plots of mixed-model logistic regression analysis (**A**). comparing severity of COVID-19 symptoms between hospitalized adult patients infected with Delta (*n* = 166) and Omicron (*n* = 905) (excluding patients with untyped VOC, *n* = 210) and (**B**). identifying risk factors for in-hospital mortality related to COVID-19 in adult patients infected with Omicron and moderate/severe/critical/fatal COVID-19 (*n* = 323) (i.e., excluding patients with asymptomatic/mild symptoms). Covariates: Omicron versus Delta; age odds increase per year; male versus female; immunocompromised versus non-immunocompromised; unvaccinated versus any vaccination; vaccinated <2 months before admission versus unvaccinated or vaccinated >2 months before admission. Abbreviations: NS, non-significant; * *p* < 0.05; ** *p* < 0.01; *** *p* < 0.001.

**Table 1 viruses-14-02736-t001:** Mixed-model logistic regression analysis comparing severity of COVID-19 outcomes between hospitalized adult patients infected with Delta (*n* = 166) and Omicron (*n* = 905) (excluding patients with untyped VOC, *n* = 210). Covariates: 1. versus Delta; 2. odds increase per year; 3. versus female; 4. versus non-immunocompromised; 5. versus any vaccination; 6. versus unvaccinated or vaccinated >2 months before admission; 7. versus unvaccinated or (1/2) basic vaccination; 8. versus vaccinated with mRNA solely. Abbreviations: OR; odds ratio; CI; confidence interval.

	Crude OR	95% CI	*p* value	Adjusted OR	95% CI	*p* Value
Outcome 1—disease severity: odds of moderate/severe/critical/fatal COVID-19 (*n* = 449/1071, 41.9%)						
Omicron ^1^	0.17	0.11–0.26	<0.001	0.14	0.09–0.22	<0.001
Age at admission ^2^	1.01	1.01–1.02	<0.001	1.02	1.01–1.03	<0.001
Male ^3^	2.07	1.60–2.67	<0.001	1.92	1.47–2.53	<0.001
Immunocompromised ^4^	2.12	1.57–2.88	<0.001	2.06	1.48–2.86	<0.001
Unvaccinated ^5^	1.54	1.09–2.16	0.01	-	-	-
Vaccinated <2 months before admission ^6^	0.55	0.38–0.78	0.001	-	-	-
Boosted ^7^	0.82	0.64–1.07	0.14	-	-	-
Viral vector vaccination ^8^	1.35	0.99–1.85	0.06	-	-	-
Outcome 2—ICU admission: admission or transferred to ICU due to COVID-19 (*n* = 112/1071, 10.5%)						
Omicron ^1^	0.22	0.14–0.34	<0.001	0.26	0.17–0.40	<0.001
Age at admission ^2^	0.99	0.98–1.00	0.01	0.99	0.98–1.00	0.02
Male ^3^	1.97	1.32–2.98	0.001	1.99	1.31–3.05	0.001
Immunocompromised ^4^	2.59	1.68–3.89	<0.01	2.34	1.50–3.56	<0.001
Unvaccinated ^5^	1.80	1.10–2.87	0.02	-	-	-
Vaccinated <2 months before admission ^6^	0.75	0.38–1.36	0.36	-	-	-
Outcome 3—mortality: odds of in-hospital mortality related to COVID-19 (*n* = 89/1071, 8.3%)						
Omicron ^1^	0.30	0.19–0.48	<0.001	0.24	0.14–0.40	<0.001
Age at admission ^2^	1.03	1.02–1.04	<0.001	1.05	1.03–1.07	<0.001
Male ^3^	2.02	1.30–3.20	0.002	1.85	1.16–2.99	0.01
Immunocompromised ^4^	2.80	1.75–4.47	<0.001	3.02	1.82–5.05	<0.001
Unvaccinated ^5^	1.15	0.56–2.19	0.68	-	-	-
Vaccinated <2 months before admission ^6^	0.51	0.15–1.34	0.22	-	-	-

## Data Availability

The data presented in this study are available on request from the corresponding author.

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
