# Peer review of "Severity of COVID-19 among Hospitalized Patients: Omicron Remains a Severe Threat for Immunocompromised Hosts"

_viruses, 2022, doi:10.3390/v14122736_

Round 1

Reviewer 1 Report

Overall, this study is relevant because the burden of Omicron disease to immunocompromised patients in general is not very well defined for inpatients nor for outpatients. At least a subgroup of immcompr patients is likely at continued risk for a poor outcome. This study focuses on admitted patients with or because of COVIC in delta and omicron era and the authors show an increased risk for a poor outcome in immunocompr patients in the omicron era. The strengths are the relatively large sample size (although not for the delta subgroup, and not for the booster vaccinated omicron subgroup).

Here is my feedback:

Abstract:

The last sentence does not really add much.

Introd: Line 93-4. This is outdated. There are now many publications that confirm the very much reduced hospital admission rate. So please update

Line 122: I think that all hospitalized patients in these centers were tested for SARS2 regardless of symptoms. This is not the policy accross the globe/Europe. So please provide information on the testing policies in the hospitals here

Results :

Limitation of the study in general:

Number of comorbidities was not analysed. Therefore, the immune status in se could not be analysed as it is to be expected that immunocompromised patients will also have more comorbities (eg. diabetes due to predn use, renal insufficiency in kidney transplant patients etc)

To increase power of the study it would make sense to consider infections to be omicron if the variant was uknown when patients were diagnosed >2wks after omicron became more than 95% dominant in Belgium (fig1 and table 1 excludes unknown VOC patients as well which is unfortunate if these are mainly patients well after dominance of omicron in Belgium)

Other feedback:

The authors should provide the number of patients in each of the 5 subgroups of immunocompr patients as per CDC definition because there is a very important difference between cancer patients and e.g. patients after organ transplant on triple immunosuppr drugs.

Line 230-32 regarding delta subgroup of immcompr patients and outcome: please add ... but this analysis was limited by its small sample size and therefore limited statistical power.

Line 251-3: Unless you disagree with reason, please add something like this (and same for line 261-3): However, the number of symptomatic Omicron infected patients that already had received a booster vaccination at 17% was very limited. Therefore, this analysis has very limited statistical power.

Line 278: I would add “not only in the outpatient setting but also once patients require hospital admission”

Line 293: You should state the lack of sample size./power stronger because a booster is needed to be optimally protected against omicron and with only 17% of omicron infected patients having received a booster this analysis is very underpowered. 

Line 305-6: Again, please mention lack of power as well here for delta and immunocompr patients analyses as illustrated by the very wide CI in figure 1. Also very important to state is the fact that the CDC immunocompr classification also includes chemotherapy for cancer. However, several large studies have now shown that the large majority of cancer patients have a good antibody response to COVID vaccination. Therefore I suggest to state that the CDC criteria are very broad and include patients receiving chemoth for cancer and that your study was too small to look into subgroups of patientst that were more severily immunocompr (eg. triple immunosuppr therapy, e.g. anti CD20 therapy versus solid tumor cancer or low dose MTX or low dose predn

Line 364: add as limitation: Finally, immunocompromised patients very often have additional comorbidities that put them at risk for a poor COVID outcome. E.g. diabetes more often in those on corticosteroids, renal insuff more often present in organ transplant patients, etc. This is a clear limitation and should be addressed because the exact attributable risk of being immunocompromised in se may therefore be much smaller.

Figure1 : Legend regarding * ** *** is missing

Reviewer 2 Report

This work is meaningful for the treatment of COVID-19, and will be of great interest to the readers of this journal. There are some questions about statistical analyses below, which need to be improved.

1.     Since the topic is about immunocompromised hosts, I suggest the authors clearly explain the definitions of immunocompromised status and non-immunocompromised status in the method part, rather than in the supplemental section.

2.     Why were adjusted ORs not calculated  for unvaccinated, vaccinated <2 months before admission, boosted vaccine, and viral vector vaccination in Table 1 and the supplemental tables?

3.     As seen in Figure1A, vaccinated<2 months before admission is a risk factor that has a significant effect on the Omicron group. Why is this risk factor not included in Figure 1B?

4.     In Supplemental Table 3, age and sex are not significantly associated with outcome 1, while age is also not associated with outcome 2, nor was sex associated with outcome 3. Why correct the data for age and sex (line 231)?  

5.     Supplementary Figure 3: It is confusing why the widths of the bars are not the same. Since this figure express the same content as Supplementary Table 6 and only two factors have significant differences, I suggest deleting this figure.
